# Synthesis of substituted pyridines with diverse functional groups via the remodeling of (Aza) indole/Benzofuran skeletons

Kannan Vaithegi[1,2], Sihyeong Yi [1,2], Ji Hyae Lee [1], Begur Vasanthkumar Varun[1] & Seung Bum Park [1✉]

Substituted pyridines with diverse functional groups are important structural motifs found in numerous bioactive molecules. Several methodologies for the introduction of various bio-relevant functional groups to pyridine have been reported, but there is still a need for a single robust method allowing the selective introduction of multiple functional groups. This study reports a ring cleavage methodology reaction for the synthesis of 2-alkyl/aryl 3-electron-withdrawing groups (esters, sulfones, and phosphonates) 5-aminoaryl/phenol pyridines via the remodeling of 3-formyl (aza)indoles/benzofurans. Totally ninety-three 5-aminoaryl pyridines and thirty-three 5-phenol pyridines were synthesized showing the robustness of the developed methodology. The application of this methodology further provided a privileged pyridine scaffold containing biologically relevant molecules and direct drug/natural product conjugation with ethyl 2-methyl nicotinate.

[1] CRI Center for Chemical Proteomics, Department of Chemistry, Seoul National University, Seoul 08826, Republic of Korea. [2]These authors contributed equally: Kannan Vaithegi, Sihyeong Yi. ✉email: sbpark@snu.ac.kr

Pyridine is a simple six-membered heterocyclic scaffold found in various natural products, drug molecules, vitamins, and materials (Fig. 1a)[1–12]. The biological activities and physical properties of pyridine analogs can be improved by introducing various functional groups into the pyridine scaffold. For example, vitamin B3, also known as nicotinic acid and with multiple biological activities, contains the carboxylic acid moiety at the C-3 position of the pyridine[13–15]. Furthermore, di- and tri-substituted pyridines are frequently found in numerous drug molecules, natural products, and agrochemicals, including pyridoxine, epibatidine, fusaric acid, nicoboxil, vismodegib, phenoxynicotinamide, anabasamine, and clonixin[10–12]. In particular, pyridyl sulfones are widely present in diverse bioactive molecules[16,17] showing anti-inflammatory and anti-viral activities[18–20]. Pyridyl phosphonates are also valuable in the field of medicinal chemistry. Specifically, PAK-104P is a pyridyl phosphonate that alleviates drug resistance to paclitaxel and doxorubicin[21]. Therefore, the development of a robust synthetic route enabling the incorporation of sulfone and phosphonate moieties on the pyridine scaffold is highly needed in medicinal and agricultural chemistry[22–27].

Poly-substituted pyridine moieties have been obtained by the traditional Hantzsch pyridine synthesis (Fig. 1b), Chichibabin

**Fig. 1 Overview of bioactive pyridines and hypothesis of this work. a** Bioactive natural products and drug molecules containing substituted pyridines. **b** Previous synthetic strategies for substituted pyridines. **c** Proposed synthetic strategy of substituted pyridines with diverse functional groups (esters/sulfones/phosphonates).

pyridine synthesis, Bohlmann–Rahtz pyridine synthesis, etc[28–35]. However, the introduction of electron-withdrawing groups on the pyridine moiety is still challenging. For instance, pyridyl sulfones are generally synthesized by the metal-catalyzed coupling of sulfinate salts with halopyridines[36, 37] or pyridyl boronic acids[38] (Fig. 1b). Other synthetic routes include the oxidation of sulfides[39,40], pyridine modification using sulfoxylate reagents[41] or organometallic reagents[42], and displacement reactions of the sodium salts of the corresponding sulfones with pyridyl halides[43,44]. However, these synthetic protocols require stench thiol compounds[45] and hazardous byproducts are formed. In the case of pyridyl(heteroaryl) phosphonates, a cross-coupling reaction is still the best synthetic method, but these coupling reactions require hydrophosphorous derivatives and pyridyl(heteroaryl) halide, tosylates, and boronic acids in the presence of transition metal catalysts, including palladium[46–53], nickel[54], and silver[55]. Michaelis-Arbuzov reaction[56,57] and metal-free Sandmeyer-type phosphonylation[58] are alternative synthetic protocols for the formation of aryl-phosphorous bonds. However, these methods require expensive metal catalysts and ligands under harsh conditions. All of these mentioned methods have mainly focused on the synthesis of aryl sulfones and phosphonates, and a few studies have reported the heteroaryl(pyridyl) functionalization.

In the presence of ammonium acetate, β-keto ester/sulfone/phosphonate can be transformed in situ to substituted enamines that undergo aldol-type addition to *N*-substituted (aza)indole carboxaldehydes and the subsequent ring cleavage reaction to produce substituted pyridines in conjugation with *o*-amino(hetero)aryl moieties (Fig. 1c) and our group has previously reported the synthesis of heterobiaryls *via* the ring cleavage reaction of (aza)indoles[59,60]. This study reports a single methodological approach for introducing various bioactive functional groups on the pyridine scaffold. The synthesis of *m*-aryl-conjugated *o*-substituted nicotinic esters and pyridine analogs with sulfone or phosphonate groups through the remodeling of (aza)indoles/benzofurans via ring cleavage reaction was investigated to address drawbacks and limitations of previous methods.

## Results and discussion

**Working hypothesis and plausible mechanism.** Initially, we investigated the synthesis of *m*-aminopyridyl-*o*-methyl-substituted ethyl nicotinates (**3aa**) *via* the proposed ring cleavage reaction of *N*-phenylsulfonyl 3-formyl 7-azaindole (**1a**) with ethyl acetoacetate (**2a**) as a model system. Ammonium acetate was the nitrogen source for the substituted enamines, which are the key intermediates of the (aza)indole ring cleavage reaction (Fig. 2).

From the β-keto ester(**I**) and ammonium acetate is generated substituted β-amino acrylate intermediate (**II**). Then, aldol-type condensation between the β-amino acrylate intermediate and 3-formyl (aza)indole (**III**) forms intermediate (**V**) by dehydration of the (**IV**). Sequential intramolecular cyclization (**VI**) and C-N bond cleavage generates the desired *m*-aminopyridyl-*o*-methyl-substituted ethyl nicotinates (**VII**).

**Reaction optimization and substrate scope**. The developed methodology proceeded smoothly, even in the absence of an acid catalyst. However, the yields were significantly reduced with other β-ketoesters (**2b**–**2f**). Therefore, the reaction conditions were optimized by changing various parameters (see Supplementary Table 4): *N*-Phenylsulfonyl 7-azaindole 3-carboxaldehyde (**1a**) and ethyl acetoacetate (**2a**) were heated in dichloroethane (DCE) in the presence of NH₄OAc and trifluoroacetic acid (TFA) at 120 °C for 16 h to deliver the desired *m*-aminoaryl-*o*-methyl nicotinate **3aa** in a 70% yield. Under the optimized conditions, the substrate scope of this methodology was then investigated with various β-ketoesters (**2b**–2 f; see Supplementary Fig. 7), such as alkyl and aryl β-keto esters, using *N*-phenylsulfonyl 3-formyl 7-azaindole (**1a**) and indole (**1a'**) (Fig. 3). The ring cleavage reaction of 3-formyl (aza)indoles (**1a** and **1a'**) with *n*-propyl β-ketoester (**2b**) afforded *m*-aminoaryl-*o*-propyl ethyl nicotinates (**3ab** and **3a'b**) in good yields. Isopropyl (**2c**) and cyclopropyl (**2d**) β-ketoesters were also applicable to this methodology and yielded the desired substituted nicotinates (**3ac–3a'd**) in moderate yields. In fact, cyclopropyl-substituted pyridine analogs have been extensively used in medicinal chemistry. As shown in Fig. 1a, LG100268 is an agonist of retinoid X receptor (RXR)[61]. This (aza)indole cleavage reaction was also compatible with cyclohexyl (**2e**) and phenyl (**2f**) β-ketoesters, and provided *o*-cyclohexyl and *o*-phenyl nicotinate analogs (**3ae–3a'f**) from the corresponding (aza)indoles (**1a** and **1a'**).

Next, we investigated the substitution effects of indole substrates (**1b–1k**; see Supplementary Fig. 6) in the ring cleavage reaction with ethyl acetoacetate (**2a**) and confirmed the formation of desired substituted pyridine analogs in good yields, regardless of the electronic effects of the substituents (Fig. 3). *N*-Phenylsulfonyl 3-formyl indoles containing electron-withdrawing bromo (**1b–1d**) and nitro (**1e–1g**) group at the C-4, C-5, and C-6 positions, respectively, provided the desired substituted pyridines (**3ba–3ga**) in moderate to good yields. In the case of electron-donating methoxy group (**1h–1k**), the desired substituted pyridines (**3ha–3ka**) were obtained in comparable yields. Furthermore, the substrate scope of this methodology was also examined

**Fig. 2 Working hypothesis and plausible mechanism.** Plausible mechanism of the synthesis of 2-alkyl/aryl 3- esters 5-aminoaryl pyridine based on the aldol-type reaction between 3-formyl (aza)indole and β-aminoacrylate generated from β-ketoester and ammonium acetate in situ is proposed.

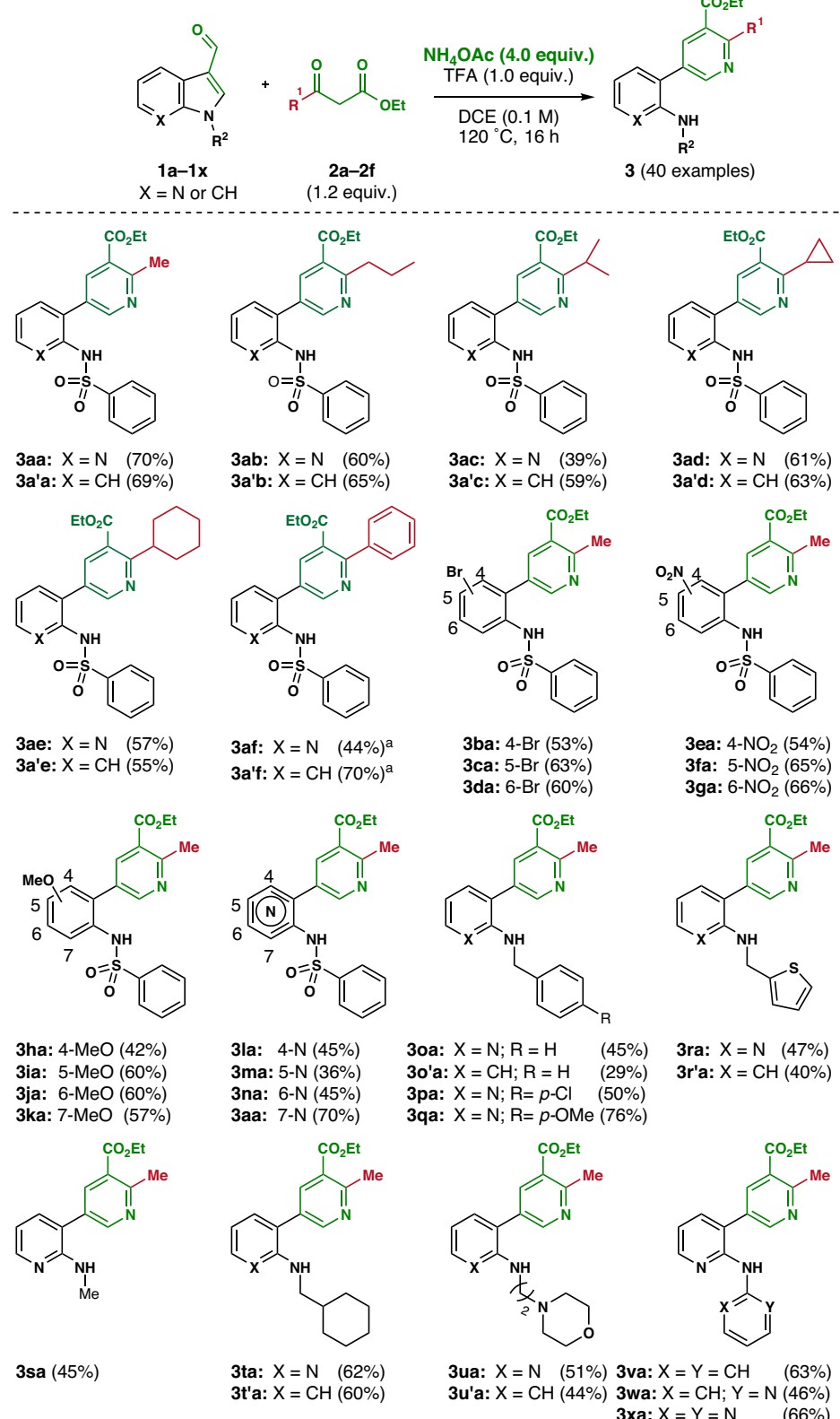

**Fig. 3 Substrate scope investigation.** Reaction conditions: **1** (0.2 mmol), **2** (1.2 equiv.), NH$_4$OAc (4.0 equiv.), TFA (1.0 equiv.) in DCE (2 mL) at 120 °C for 16 h. Yields of isolated products (**3**) are given in parenthesis. a: Reaction performed in ethanol and without trifluoroacetic acid.

using regioisomeric azaindoles (**1l–1n**) to generate diverse *o*-aminopyridyl-conjugated pyridine analogs (**3la–3na**).

The compatibility and substrate scopes of this methodology with diverse *N*-substituted (aza)indoles (**1o–1x**; see Supplementary Fig. 6) using ethyl acetoacetate (**2a**) were then investigated

(Fig. 3). 3-Formyl (aza)indoles containing *N*-benzyl substituents (**1o–1q**), regardless of the functional groups on the benzene ring, successfully proceeded the desired reaction in good to excellent yields. Furthermore, when the benzene ring was substituted with thiophene (**1r** and **1r'**), the desired transformation was well

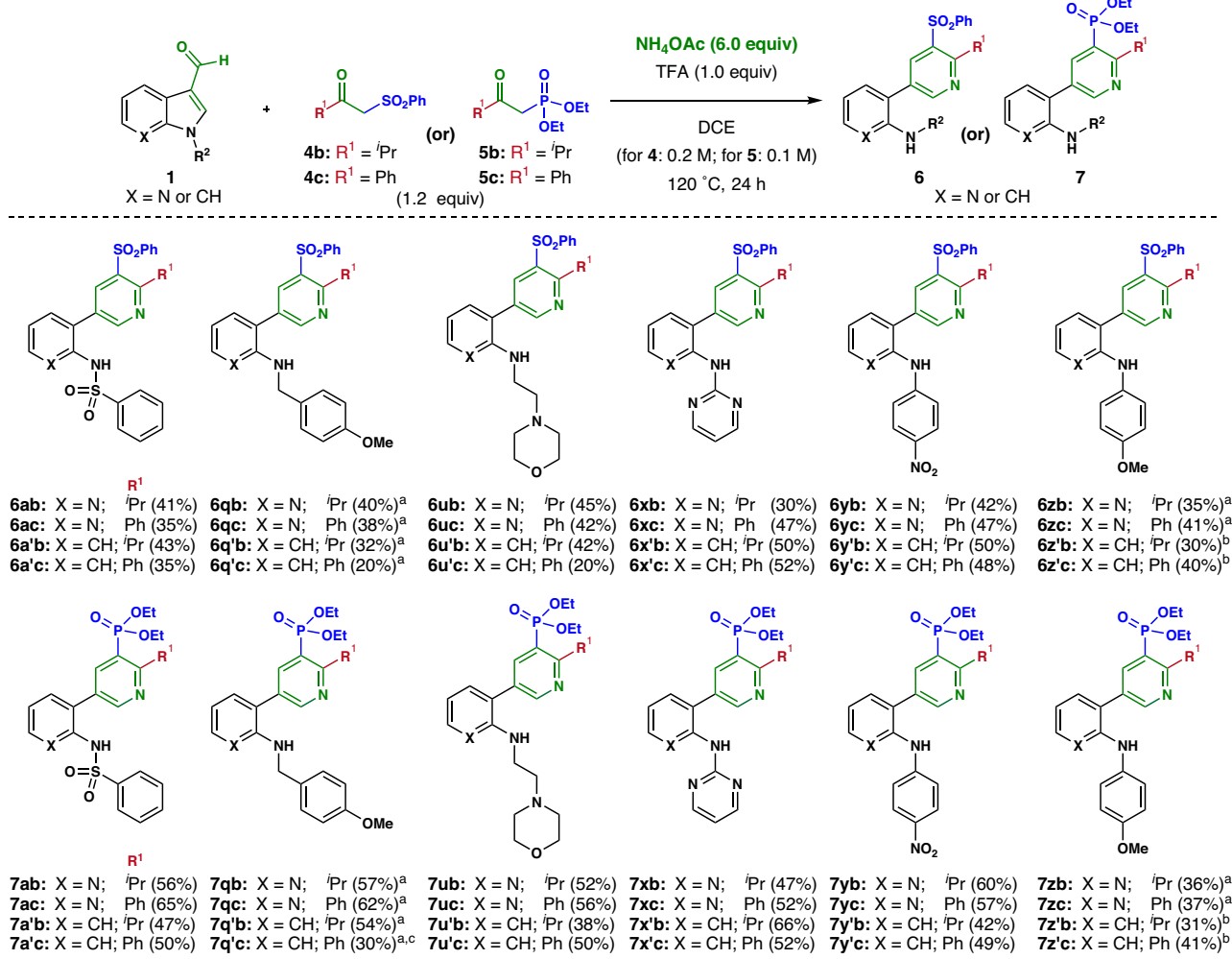

**Fig. 4 Use of various enamine sources for the synthesis of pyridyl sulfones/phosphonates.** Reaction conditions: **1** (0.2 mmol), β-keto sulfones/phosphonates (**4b–c/5b–c**, 1.2 equiv.), NH₄OAc (6.0 equiv.), TFA (1.0 equiv.) in DCE (2 mL/1 mL) at 120 °C for 16 h to 48 h. Yields of isolated products (**6** and **7**) are given in parenthesis. a: 30 h reaction time. b: 48 h reaction time. c: 55% Crude NMR yield.

achieved. 3-Formyl (aza)indoles containing alkyl groups, such as methyl (**1s**), cyclohexylmethyl (**1t** and **1t'**), 2-morpholinoethyl (**1u** and **1u'**), were also converted to the desired pyridine analogs (**3sa–3u'a**). It is worth mentioning that *N*-aryl-substituted 3-formyl azaindoles, such as phenyl (**1v**), pyridyl (**1w**), and pyrimidyl (**1x**) moieties, suited well with this ring cleavage methodology, and *N*-aryl-substituted aminopyridyl nicotinates (**3va–3xa**) were obtained in moderate to good yields.

**Use of β-keto sulfones and β-keto phosphonates**. We next examined the robust synthesis of pyridine analogs containing various sulfones and phosphonates using our methodology (Fig. 4). The reaction compatibility of *N*-substituted 3-formyl (aza)indole (**1a** and **1a'**) with phenyl sulfonyl acetone (**4a**; see Supplementary Fig. 2) was initially explored, but the undesired *o*-methylsulfonyl pyridine (**6aa'**) was formed along with the desired *o*-methyl-*m*-sulfonyl pyridine analog (**6aa**) due to the regioselectivity of enamine formation (see Supplementary Fig. 2). A similar reactivity pattern was observed in the case of diethyl (2-oxopropyl)phosphonate (**5a**; see Supplementary Fig. 3). To address this issue, we used 3,3-dimethyl phenylsulfonyl acetone (**4b**) as the source of sulfonyl enamine, and successfully obtained the desired *o*-isopropyl-*m*-sulfonyl pyridine (**6ab**) without forming its regioisomers (see Supplementary Fig. 4). Unlike β-ketoesters, the corresponding sulfones and phosphonates were not sufficiently reactive.

Therefore, we further optimized the reaction conditions, and confirmed that isopropyl (**4b**) and phenyl β-ketosulfones (**4c**) were less reactive than their phosphonate analogs (**5b** and **5c**). A higher reaction concentration was thus needed. Under the re-optimized conditions (as shown in Fig. 4), the reactivities of *N*-phenylsulfonyl 3-formyl 7-azaindole (**1a**) and indole (**1a'**) with β-keto sulfones (**4b–4c**) or phosphonates (**5b–5c**) were investigated. The desired *m*-(hetero)aryl pyridyl sulfones (**6ab–6a'c**) and phosphonates (**7ab–7a'c**) were obtained in moderate to good yields. In the case of the *N*-alkyl substituents, *N*-*p*-methoxybenzyl (**1q** and **1q'**) and *N*-(2-morpholinoethyl)-3-formyl (aza)indoles (**1u** and **1u'**) afforded the desired ring cleavage products (**6qb–6u'c** and **7qb–7u'c**) in moderate yields. This reactivity pattern was further confirmed with *N*-pyrimidyl (**1x** and **1x'**), *N*-*p*-nitrophenyl (**1y** and **1y'**), *N*-*p*-methoxyphenyl (**1z** and **1z'**) 3-formyl (aza)indoles. Compared to the electron-donating *N*-*p*-methoxyphenyl analogs (**6zb–6z'c** and **7zb–7z'c**), the electron-withdrawing *N*-pyrimidyl and *N*-*p*-nitrophenyl 3-formyl (aza)indoles provided the desired pyridyl sulfones and phosphonates in better yields (**6xb–6x'c** and **7xb–7x'c**; **6yb–6y'c** and **7yb–7y'c**).

**Use of benzofurans as substrates**. Once the general reactivity of *N*-substituted 3-formyl (aza)indoles with a series of enamines (in situ generated from β-keto esters, sulfones, and phosphonates) was confirmed for the synthesis of highly functionalized pyridines, the

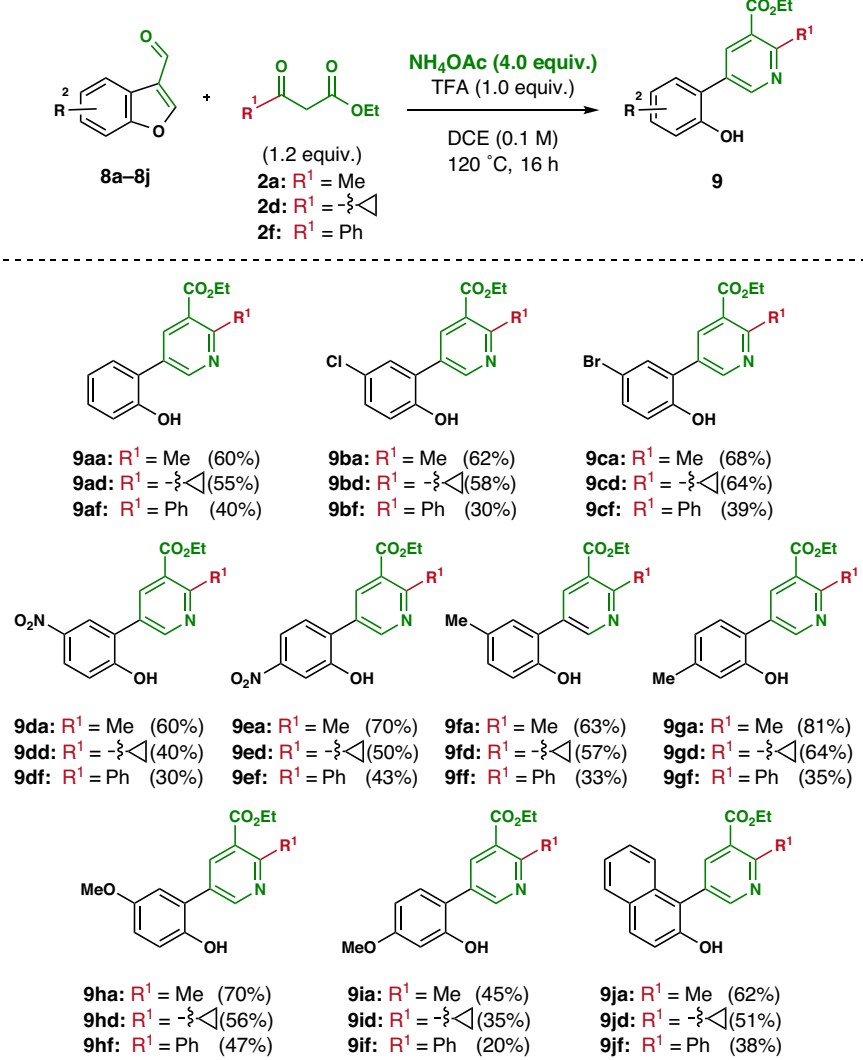

**Fig. 5 Reaction with benzofuran analogs.** Reaction conditions: **8** (0.2 mmol), $\beta$-ketoesters (**2a**, **2d**, or **2f**, 1.2 equiv.), NH$_4$OAc (4.0 equiv.), TFA (1.0 equiv.) in DCE (2 mL) at 120 °C for 16 h. Yields of isolated products (**9**) are reported.

scope of the ring cleavage methodology was extended to benzofuran derivatives. Benzofuran is an oxygen-containing heterocycle found in diverse natural products and bioactive molecules[62], but up to our knowledge the remodeling of benzofuran skeletons to *N*-heterocycles has not yet been reported. The reactivity of 3-formyl benzofurans with representative $\beta$-ketoesters using this methodology to harness *o*-substituted-*m*-phenol-conjugated nicotinates was investigated (Fig. 5). In fact, the phenol and heterobiaryl moieties are one of the most abundant structural units found in numerous bioactive natural products and therapeutic agents[63,64]. Unlike indoles and azaindoles, most of 3-formyl benzofurans are not commercially available. Therefore, the substituted 3-formyl benzofuran analogs (**8b–8j**) were prepared from their corresponding salicylaldehydes (see Supplementary Fig. 14). As a model system, methyl (**2a**), cyclopropyl (**2d**), and phenyl (**2f**) $\beta$-ketoesters were chosen as the enamine sources. Under the optimized conditions, 3-formyl benzofuran (**8a**) was reacted with three representative $\beta$-ketoesters to deliver the desired ring cleavage product containing methyl (**9aa**), cyclopropyl (**9ad**), and phenyl (**9af**) moieties at the C-2 position. We then examine the substrate scope of this ring cleavage reaction using various 5- and 6-substituted 3-formyl benzofurans (**8b–8i**) and obtained highly functionalized nicotinate derivatives (**9ba–9ia**, **9bd–9id**, and **9bf–9if**) in moderate to good yields. In particular, the reaction of 3-formyl benzofurans with both electron-withdrawing groups

(chloro, bromo, and nitro; **8b–8e**) and electron-donating groups (methyl and methoxy; **8f–8i**) at the C-5 and C-6 positions afforded the desired (hetero)biaryl products. 3-Formyl naphthofuran (**8j**) also provided the *o*-substituted nicotinate analogs containing a naphthol moiety (**9ja**, **9jd**, and **9jf**).

**Synthetic application.** This (aza)indole ring cleavage reaction was then applied to the synthesis of analogs of various bioactive pyridines. Initially, privileged structural units were extracted from bioactive natural products (fusaric acid and epibatidine) and drug molecules (clonixin, nicoboxil, and vismodegib). These privileged pyridine scaffolds were then synthesized using (aza)indole ring cleavage methodology. Highly functionalized pyridine analogs (**10a–10e**) were synthesized from *N*-substituted 3-formyl (aza) indoles (Fig. 6a) with diverse $\beta$-ketoesters/sulfones (**2a**, **2g**, **2h**, **2i**, and **4d**; see Supplementary Fig. 7). For example, *N*-aryl 3-formyl azaindole (**1aa**) was reacted with ethyl acetoacetate (**2a**) to furnish the non-steroidal anti-inflammatory drug (NSAID) clonixin analog (**10a**) in a 59% yield. Other bio-relevant molecules were synthesized in a single step from *N*-phenylsulfonyl 3-formyl indole (**1a'**) *via* the ring cleavage reaction with the corresponding $\beta$-ketoesters (**2g–2i**) to afford the nicoboxil analog (**10b**), the fusaric acid analog (**10c**), and the epibatidine analog (**10d**) in 78%, 60%, and 62% yields,

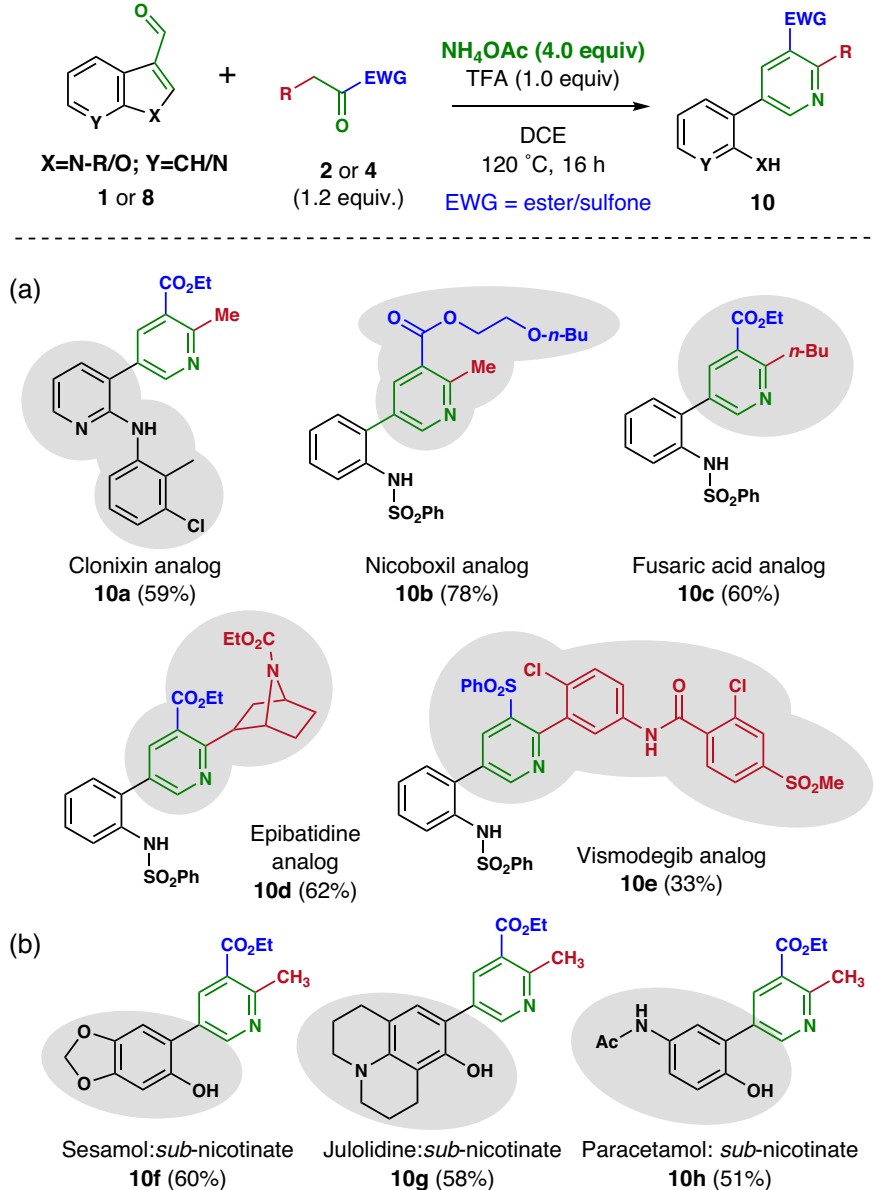

**Fig. 6 Synthetic application. a** Synthesis of privileged pyridine scaffold-containing bio-relevant molecules. **b** Drug/natural product conjugation with ethyl 2-methyl nicotinate. See Supplementary information for reaction condition and further details.

respectively. Nicoboxil is an FDA-approved drug used for the treatment of acute back pain. Fusaric acid and epibatidine are pyridine scaffold-containing natural products whereby fusaric acid is an antibiotic isolated from the fungus *Fusarium heterosporium* and used for the synthesis of vasodilator bupicomide[65], while epibatidine is known as a modulator of the nicotinic acetylcholine receptor[66, 67]. The ring cleavage reaction of *N*-phenylsulfonyl 3-formyl indole (**1a'**) with aryl β-ketosulfone (**4d**) also furnished the desired vismodegib analog (**10e**). Vismodegib is an FDA-approved drug used for the treatment of basal cell carcinoma.

Consequently, we applied this late-stage transformation method to the field of drug-drug or drug-natural product conjugation (Fig. 6b). Substituted 3-formyl benzofurans (**8k** and **8l**) were reacted with ethyl acetoacetate (**2a**), which allowed the formation of sesamol-conjugated 2-methyl nicotinate (**10g**, natural product-drug conjugate) and julolidine-nicotinate conjugate (**10h**) in 60% and 58% yields, respectively. Naturally occurring sesamol exhibits anti-fungal activity and can be used to synthesize paroxetine (sold under the brand names Paxil® and

Seroxat®), a type of antidepressant drug under the class of selective serotonin reuptake inhibitors (SSRI). Julolidine is a heterocyclic aromatic moiety extensively used in therapeutic agents, photoconductive materials, and chemiluminescence substances. Paracetamol-conjugated 2-methyl nicotinate (**10f**, drug-drug conjugate) was also obtained from the corresponding 3-formyl benzofuran (**8m**) in a 51% yield. Paracetamol, or acetaminophen, is used as an analgesic and antipyretic. These examples demonstrated that the proposed ring cleavage reaction could be beneficial for synthesizing highly functionalized privileged pyridines *via* the late-stage remodeling of (aza)indoles and benzofurans.

## Conclusions

In conclusion, this study reported the successful application of the proposed ring cleavage strategy for the synthesis of *o*-substituted *m*-aminoaryl-conjugated pyridines from *N*-substituted 3-formyl (aza)indoles. In fact, this reaction afforded diversely substituted

pyridine analogs containing multiple functional groups, such as esters, sulfones, and phosphonates, at the C-3 position of pyridine with a wide range of substrate scope, which is not easily accessible by conventional methods. Furthermore, this ring cleavage reaction was extended to benzofuran derivatives for synthesizing *m*-phenol-conjugated (hetero)biaryl nicotinates. Though *o*-aniline/phenol were inevitably incorporated on the *m*-position of pyridine, these moieties can enhance the bio-relevancy of the final biaryl structures due to their abundancy in drugs and bioactive molecules. synthetic methodology allowed access to various analogs of drugs and biologically relevant molecules containing privileged pyridine scaffolds. Finally, this methodology allowed the late-stage conjugation of substituted nicotinates with paracetamol, sesamol, and julolidine as drug-drug and natural product-drug conjugates from 3-formyl benzofurans. Biological studies on all the synthesized compounds are currently in progress, and the outcomes will be reported in due course.

## Methods

**General methods.** For instrumentation and materials, see Supplementary Method - General Information. For Additional experiments concerning optimization of the reaction conditions, see Supplementary Figures – (2) Reaction Optimization.

**General procedure for the reaction of *N*-substituted 3-formyl (aza)indoles with diverse *β*-ketoesters (2a–2f).** A 4-mL vial equipped with a magnetic bar and a Teflon-lined screwed cap was charged with **1** (0.2 mmol), *β*-ketoesters (**2a–2f**, 1.2 equiv.), trifluoroacetic acid (TFA, 22.80 mg, 14.86 μL, 1.0 equiv.), and $NH_4OAc$ (61.66 mg, 4.0 equiv.) in dichloroethane (DCE, 2.0 mL). The vial was then sealed and heated at 120 °C for 16 h. Upon reaction completion checked by LC-MS and TLC analysis, the reaction mixture was diluted with dichloromethane (DCM), quenched with saturated aqueous $NaHCO_3$ solution, and extracted with DCM (3 × 10 mL). The combined organic layer was washed with brine (10 mL), dried over anhydrous $Na_2SO_4$(s), filtered, and concentrated under reduced pressure. The crude mixture was purified by silica-gel flash column chromatography to obtain the desired product.

Note: The general procedure for the above methodology was slightly modified in the case of ethyl benzoylacetate (**2f**); the reaction was performed in ethanol without TFA.

**General procedure for the reaction of *N*-substituted 3-formyl (aza)indole with *β*-ketosulfones (4b–c)/*β*-ketophosphonates (5b–c).** A 4-mL vial equipped with a magnetic bar and a Teflon-lined screwed cap was charged with **1** (0.2 mmol), *β*-keto sulfones/phosphonates (**4b–c/5b–c**, 1.2 equiv.), TFA (22.80 mg, 14.86 μL, 1.0 equiv.), and $NH_4OAc$ (92.50 mg, 6.0 equiv.) in DCE (1.0 mL (**4b–c**)/2.0 mL (**5b–c**)). The vial was then sealed and heated at 120 °C for 16 h to 48 h. Upon reaction completion checked by LC-MS and TLC analysis, the reaction mixture was diluted with DCM, quenched with saturated aqueous $NaHCO_3$ solution, and extracted with DCM (3 × 10 mL). The combined organic fraction was washed with brine (10 mL), dried over anhydrous $Na_2SO_4$(s), filtered, and concentrated under reduced pressure. The crude compound was purified by silica-gel flash column chromatography to obtain the desired product bearing 3-pyridylsulfones (**6**)/3-pyridyl phosphonates (**7**).

**General procedure for the reaction of benzofuran-3-carboxaldehydes with *β*-ketoesters.** A 4-mL vial equipped with a magnetic bar and a Teflon-lined screwed cap was charged with **8** (0.2 mmol), *β*-ketoesters (**2a, 2d**, or **2f**, 1.2 equiv.), TFA (22.80 mg, 14.86 μL, 1.0 equiv.), and $NH_4OAc$ (61.66 mg, 4.0 equiv.) in DCE (2.0 mL). The vial was then sealed and heated at 120 °C for 16 h. Upon reaction completion checked by LC-MS and TLC analysis, the reaction mixture was diluted with DCM, quenched with saturated aqueous $NaHCO_3$ solution, and extracted with DCM (3 × 10 mL). The combined organic fraction was washed with brine (10 mL), dried over anhydrous $Na_2SO_4$(s), filtered, and concentrated under reduced pressure. The crude compound was purified by silica-gel flash column chromatography to obtain the desired phenol-conjugated product (**9**).

**Preparation of substrates.** See Supplementary Method - Supplementary Figs. 9–22.

**Spectroscopic data of products.** See Supplementary Data 1.

## Data availability

All data generated and analyzed during this study are included in this article, its Supplementary Information, and Supplementary Data, and also available from the authors upon reasonable request.

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

## Acknowledgements

This research was funded by National Creative Research Initiative Grant (2014R1A3A2030423) through the National Research Foundation of Korea (NRF) funded by the Korean Government (Ministry of Science & ICT). Yi, S. and Lee, J.H. are grateful for the fellowship by BK21 Plus Program.

## Author contributions

K.V. and S.Y. designed and optimized the methodology, performed synthetic experiments, characterized compounds, and prepared the manuscript. J.H.L. and B.V.V. performed the synthesis and characterization. S.B.P. directed the whole study and involved in all aspects of the experimental design, data analysis, and manuscript preparation. All authors critically reviewed the text and figures.

## Competing interests

The authors declare no competing interests.
