## [Peer Review File · Communications Chemistry]

Synthesis of Substituted Pyridines with Diverse Functional Groups via the Remodeling of (Aza)indole/Benzofuran SkeletonsReviewers' comments:

Reviewer #1 (Remarks to the Author):

This manuscript by Park and co-workers reports a versatile synthetic protocol of 2, 3, 5-trisubstituted pyridine derivatives featuring a ring cleavage approach of (aza)indoles/benzofurans. Based on the closely related protocols exploiting the manipulation of 3-formyl (aza)indoles to effect formation of pyridines followed by a C-N bond cleavage of the (aza)indole scaffold (ref. 59: *Org. Lett.* 2009, 11, 5214. and ref. 60: *Nat. Commun.* 2020, 11, 6308.), the authors developed more widely expandable approaches featuring assembly of the three components, (aza)indoles/benzofurans, β -keto carbonyls/sulfones/phosphonates, and ammonium acetate. Whilst the previously reported protocols rely on either aminopyrazoles (ref. 59) or propiolates (ref. 60) as building blocks, the current approach efficiently exploits a series of enamines generated in situ from ammonium acetate and β -keto carbonyls/sulfones/phosphonates. The authors also demonstrated applicability of benzofurans in place of the (aza)indoles under essentially the same conditions, which allowed successful development of a highly generalizable synthetic protocol. Furthermore, this paper reports synthetic application to generate five drug analogs (Fig 5a) and three hybrid molecules bearing both trisubstituted pyridines moiety and natural product-relevant phenols (Fig 5b). Thus, the authors efficiently developed a unified and versatile synthetic protocol to gain systematic access to various analogs of drugs and biologically intriguing molecules bearing the privileged pyridine scaffolds. This reviewer would like to recommend publication of this manuscript in *Commun. Chem* after minor revision.

- (1) The plausible reaction mechanism (Supplementary Figure 1) should be provided and concisely explained in the main text to facilitate understanding for readers.
- (2) Fig. 5: Illustration of reaction schemes indicating structures of the two substrates, pre-functionalized (aza)indoles/benzofurans and corresponding β -keto carbonyls/sulfones, would be beneficial for readers. The use of colors to highlight bonds should be consistent.
- (3) Fig. 1, SI Fig S1: $P(O)OEt_2 \rightarrow PO(OEt)_2$
- (4) Fig. 2 top line: The bond length between R1 (red) and C=O (green) should be a bit longer.
- (5) The information of ref. 60 should be corrected (delete 2020 111).

Reviewer #2 (Remarks to the Author):

"Synthesis of Substituted Pyridines with Diverse Functional Groups via the Remodeling of (Aza)indole/Benzofuran Skeletons"

Overview

The authors report a method to construct substituted pyridines from formylated indole and benzofurans (and derivatives) with β -keto esters, β -keto sulfones and β -keto phosphonates, and ammonium acetate as reagents. They make the case that substituted pyridines are important due to their effect on biological systems and briefly outline other methods to make these types of structures. Their approach involves a "remodeling" strategy where a condensation process precedes a ring opening step within the

indole or benzofuran. The substituent on the keto portion of the coupling partner ends up at the 2-position, and then an ester, sulfone, or phosphonate group resides at the 3-position of the pyridine ring. They present the scope of this process in Figures 2-4 and then make derivatives of recognized bioactive compounds in Figure 5.

Critique

There are two central issues that I will address when evaluating this paper:

1) Value of the substitute pyridines. The authors are correct that many examples of substituted pyridines in bioactive molecules exist. My contention is that the product pyridines are in a tiny subset of structures and are not likely to find significant utility. Each pyridine is tris-substituted, which is perfectly fine; the problem is that they possess an aryl group at the 2-position with an ortho-heteroatom substituent. The chances of practitioners requiring this motif are very small, which diminishes this approach's utility. The area of 'molecular editing' or 'skeletal remodeling' is a hot topic right now. However, this attention does not mean any reaction under that umbrella is useful. Incorporating alkyl and aryl groups and the 2-position and esters, sulphones, and phosphonate esters at the 3-position is much better. It is a shame that there is no way to avoid the more esoteric aryl group, as the reaction mechanism is intrinsically tied to this moiety.

2) Nature of the starting materials and importance of the process as a strategic method for pyridine synthesis. The β -keto esters, β -keto sulfones, and β -keto phosphonates are useful as starting materials as they are abundant and straightforward to synthesize. The formylated indoles and benzofurans are perhaps more of an issue as they are not abundant, and because of the issue described above with the heteroatom substituent residing in the products at the end of the process. To be fair, the authors have generated a reasonable range of products via this approach, but I don't think it will be something that practitioners will exploit unless they are interested in this specific substrate class.

Considering points 1 and 2 above, I would place this paper as a borderline case for Communications Chemistry. At first glance, the pyridines look somewhat relevant but fall into a niche category, as described in point 1. The chemistry would be much more impactful without this restriction and improve if the authors showed that this group could be conveniently removed and/or modified.

For Reviewer #1.

This manuscript by Park and co-workers reports a versatile synthetic protocol of 2, 3, 5-trisubstituted pyridine derivatives featuring a ring cleavage approach of (aza)indoles/benzofurans. Based on the closely related protocols exploiting the manipulation of 3-formyl (aza)indoles to effect formation of pyridines followed by a C-N bond cleavage of the (aza)indole scaffold (ref. 59: *Org. Lett.* 2009, 11, 5214. and ref. 60: *Nat. Commun.* 2020, 11, 6308.), the authors developed more widely expandable approaches featuring assembly of the three components, (aza)indoles/benzofurans, β -keto carbonyls/sulfones/phosphonates, and ammonium acetate. Whilst the previously reported protocols rely on either aminopyrazoles (ref. 59) or propiolates (ref. 60) as building blocks, the current approach efficiently exploits a series of enamines generated in situ from ammonium acetate and β -keto carbonyls/sulfones/phosphonates. The authors also demonstrated applicability of benzofurans in place of the (aza)indoles under essentially the same conditions, which allowed successful development of a highly generalizable synthetic protocol. Furthermore, this paper reports synthetic application to generate five drug analogs (Fig 5a) and three hybrid molecules bearing both trisubstituted pyridines moiety and natural product-relevant phenols (Fig 5b). Thus, the authors efficiently developed a unified and versatile synthetic protocol to gain systematic access to various analogs of drugs and biologically intriguing molecules bearing the privileged pyridine scaffolds. This reviewer would like to recommend publication of this manuscript in *Commun. Chem* after minor revision.

We sincerely appreciate the kind recommendation of publication and thoughtful comments by Reviewer 1. We modified our manuscript and supporting information based on your suggestion, which made our work more straightforward and informative. All the changes are now highlighted in our modified manuscript, which will be helpful for you to see the changes. Particular answers for point-by-point are listed in the following.

(1) The plausible reaction mechanism (Supplementary Figure 1) should be provided and concisely explained in the main text to facilitate understanding for readers.

We appreciate your thoughtful comments. Based on your suggestion, we added the plausible reaction mechanism in Figure 2 as the “Working hypothesis and plausible mechanism” section at the beginning of the results and discussion. Figure numbers of later figures were modified according to the sequence in the revised manuscript.

Figure 2 (new) and the following paragraph were added in the revised manuscript (page 2) to explain the plausible mechanism of this transformation concisely.

Fig. 2 Working hypothesis and plausible mechanism

“Working hypothesis and plausible mechanism. Initially, we investigated the synthesis of *m*-aminopyridyl-*o*-methyl-substituted ethyl nicotines (**3aa**) via the proposed ring cleavage reaction of *N*-phenylsulfonyl 3-formyl 7-azaindole (**1a**) with ethyl acetoacetate (**2a**) as a model system. Ammonium acetate was the nitrogen source for the substituted enamines, which are the key intermediates of the (aza)indole ring cleavage reaction (Fig. 2). From the β -keto ester(I) and ammonium acetate is generated β -amino acrylate intermediate (II). Then, aldol-type condensation between the β -amino acrylate intermediate and 3-formyl (aza)indole (III) forms intermediate (V) by dehydration of the (IV). Sequential intramolecular cyclization (VI) and C-N bond cleavage generate the desired *m*-aminopyridyl-*o*-methyl-substituted ethyl nicotines (VII).”

(2) Fig. 5: Illustration of reaction schemes indicating structures of the two substrates, pre-functionalized (aza)indoles/benzofurans and corresponding β -keto carbonyls/sulfones, would be beneficial for readers. The use of colors to highlight bonds should be consistent.

Based on your suggestion, we added the general scheme on the above of the structures of synthetic application (Figure 6 in the revised manuscript). In addition, we colored EWG in blue and *ortho*-substituents in red with the consistency of previous figures to improve the clarity, which benefit general readers' understanding.

Fig. 6 Synthetic application. a Synthesis of privileged pyridine scaffold-containing bio-relevant molecules. **b** Drug/natural product conjugation with ethyl 2-methyl nicotinate. See Supplementary information for reaction conditions and further details.

(3) Fig. 1, SI Fig S1: $P(O)OEt_2 \rightarrow PO(OEt)_2$

We revised Figure 1 and Supplementary Figure 1 according to your comments.

(4) Fig. 2 top line: The bond length between R1(red) and C=O (green) should be a bit longer.

In fact, the bond length settings are identical, but due to the location of superscripts, the bond looks shorter. We moved the superscript to make the bond more visible. Thank you for your thoughtful comments.

(5) The information of ref. 60 should be corrected (delete 2020 111).

We re-checked and corrected the reference, including ref.60 in the revised manuscript.

For Reviewer #2.

"Synthesis of Substituted Pyridines with Diverse Functional Groups via the Remodeling of (Aza)indole/Benzofuran Skeletons"

Overview

The authors report a method to construct substituted pyridines from formylated indole and benzofurans (and derivatives) with β -keto esters, β -keto sulfones and β -keto phosphonates, and ammonium acetate as reagents. They make the case that substituted pyridines are important due to their effect on biological systems and briefly outline other methods to make these types of structures. Their approach involves a "remodeling" strategy where a condensation process precedes a ring opening step within the indole or benzofuran. The substituent on the keto portion of the coupling partner ends up at the 2-position, and then an ester, sulfone, or phosphonate group resides at the 3-position of the pyridine ring. They present the scope of this process in Figures 2-4 and then make derivatives of recognized bioactive compounds in Figure 5.

We sincerely appreciate the thoughtful and critical comments by Reviewer 2. We could get new insights regarding our work from your critical review. Though your comments and directions are not fully incorporated within this work, we hope we can develop excellent future work regarding your critical insight. We prepared our answers for your critiques in the following.

Critique

There are two central issues that I will address when evaluating this paper:

1) Value of the substitute pyridines. The authors are correct that many examples of substituted pyridines in bioactive molecules exist. My contention is that the product pyridines are in a tiny subset of structures and are not likely to find significant utility. Each pyridine is tris-substituted, which is perfectly fine; the problem is that they possess an aryl group at the 2-position with an ortho-heteroatom substituent. The chances of practitioners requiring this motif are very small, which diminishes this approach's utility. The area of 'molecular editing' or 'skeletal remodeling' is a hot topic right now. However, this attention does not mean any reaction under that umbrella is useful. Incorporating alkyl and aryl groups and the 2-position and esters, sulphones, and phosphonate esters at the 3-position is much better. It is a shame that there is no way to avoid the more esoteric aryl group, as the reaction mechanism is intrinsically tied to this moiety.

We appreciate the thoughtful comment and insight about this study. We do agree that our methodology allows access to substituted pyridines with aminoaryl/phenol moieties at the 5-position. Our methodology might not be attractive to those who want to generate pyridine itself. There are multiple different methods to synthesize the pyridine analogs. However, we believe that incorporating aminoaryl/phenol into the molecule can enhance bio-relevancy. It is clear that there is fewer examples of these type of structures, but it doesn't mean that these structures are less important. These examples are rare since *ortho*-heteroatom-containing biaryl molecules are not accessible or difficult to access *via* conventional transition metal-mediated aryl-aryl coupling methodologies.

In addition, anilines and phenols are structural motives widely found in bioactive natural products and therapeutic agents, as mentioned in our manuscript. More than 100 drug molecules can be found in *Drug Bank Database*,¹ which has aniline or phenol for each. Therefore, incorporating aniline or phenol in the molecular framework can be beneficial to ensure biological relevancy. Moreover, biaryl moiety itself is one of the privileged structures, and the robust synthetic method for heterobiaryl moiety can maximize the bio-relevancy of the drug-like molecules.

According to the *Drug Bank Database*, 55 bioactive molecules possess *ortho*-phenol biaryl structures. Well-known antibiotic, Vancomycin itself, has this moiety. Especially in many RiPPs and tyrosine-containing cyclic peptides possess *ortho*-phenol biaryl structure. 5 bioactive *ortho*-aniline biaryl structures were also found in the *Drug Bank Database*, and among them two were approved; one is *boscalid* and the other is *revefenacin*. When we extended the aniline to aminopyridines, 13 more bioactive compounds were found (five 2-aminopyridines, one 3-aminopyridines, two 4-aminopyridine, and five 5-aminopyridines). Based on these statistics, we can claim that connecting an *ortho*-heteroaryl ring to pyridine structure can be rare and not easily accessible, but *ortho*-heterobiaryl ring containing pyridine itself can be an important class of potential drugs.

In this study, we aimed to generate a new methodology for the synthesis of diverse pyridines with high biological relevancy and successfully demonstrated the synthesis of a series of heterobiaryl moieties with various substituents containing *ortho*-aniline/phenol as a part of bio-relevant functionalities *via* skeletal remodeling of (aza)indoles. Therefore, we added the following sentence admitting that the incorporation of *o*-aniline/phenol was inevitable in the conclusion part of the revised manuscript.

“Though o-aniline/phenol was inevitably incorporated at the m-position of pyridine, these moieties can enhance the bio-relevancy of the final biaryl structures due to their abundance in drugs and bioactive molecules.”

2) Nature of the starting materials and importance of the process as a strategic method for pyridine synthesis. The β -keto esters, β -keto sulfones, and β -keto phosphonates are useful as starting materials as they are abundant and straightforward to synthesize. The formylated indoles and benzofurans are perhaps more of an issue as they are not abundant, and because of the issue described above with the heteroatom substituent residing in the products at the end of the process. To be fair, the authors have generated a reasonable range of products via this approach, but I don't think it will be something that practitioners will exploit unless they are interested in this specific substrate class.

We appreciate the thoughtful comments by reviewer 2. We agree that the starting material's nature is important in methodology. We are also grateful that you agree with the abundance and straightforwardness of β -keto substrates.

¹ <https://go.drugbank.com>

However, we are afraid that the abundance and straightforwardness of formylated indoles seem to be underestimated. Due to the importance of (aza)indole structure, the side residue of tryptophan, (aza)indole moiety is widely found in bioactive natural products. Thus, plenty of (aza)indole structures are commercially available. In addition, the Vilsmeier-Haack method for indoles² and the Duff method³ for azaindoles are well established for the formylation process, thanks to the development of various tryptamine-based synthetic procedures. In this study, we have 93 examples using 3-formyl (aza)indoles, which strongly support the feasibility of this transformation.

In the case of benzofurans, they are relatively less commercially available. However, as shown in our supporting information, especially in supplementary Figure 14, all the 3-formyl benzofurans were synthesized from diverse commercially available salicylaldehydes with simple and straightforward 3-step reactions. No rare reagents nor environment-sensitive methods were used.

In summary, we believe that our starting materials are widely-being-used materials by the practitioners. The collection of examples we've shown in this study can prove the practicality of the starting materials by itself.

Considering points 1 and 2 above, I would place this paper as a borderline case for Communications Chemistry. At first glance, the pyridines look somewhat relevant but fall into a niche category, as described in point 1. The chemistry would be much more impactful without this restriction and improvement if the authors showed that this group could be conveniently removed and/or modified.

² *J. Am. Chem. Soc.* **1952**, *74*, 2273; *J. Chem. Soc.* **1954**, 3842-3846

³ *J. Chem. Soc.* **1941**, 547; *J. Chem. Soc.* **1945**, 276; *J. Chem. Soc.* **1951**, 1512.